# Indonesian Science, Mathematics, and Engineering Preservice Teachers' Experiences in STEM-TPACK Design-Based Learning

**Ching Sing Chai [1], Yuli Rahmawati [2],\* and Morris Siu-Yung Jong [1],\***

1   Department of Curriculum and Instruction & Centre for Learning Sciences and Technologies, The Chinese University of Hong Kong, Shatin, Hong Kong; CSChai@cuhk.edu.hk

2   Department of Chemistry Education, Universitas Negeri Jakarta, Jakarta 13320, Indonesia

\*   Correspondence: yrahmawati@unj.ac.id (Y.R.); mjong@cuhk.edu.hk (M.S.-Y.J.)

**Abstract:**   This paper presents Indonesian preservice teachers' experiences in designing a Science Technology Engineering and Mathematics-Technological Pedagogical Content Knowledge (STEM-TPACK) learning website. The framework of TPACK was expanded to include all STEM subjects for the synthesis of the theoretical/design framework. The STEM-TPACK framework is further epitomized as a replicable website to support preservice teachers in designing STEM lesson activities. The framework is also employed to examine preservice teachers' efficacies and experiences in learning how to design the learning website. Thirty-seven second- and third-year Indonesian preservice teachers from science, mathematics, computer science, and engineering backgrounds formed interdisciplinary groups to design the STEM-TPACK website based on the current secondary school curricula. Data were collected from TPACK-STEM questionnaires, interviews, reflective journals, and observation. The preservice teachers' efficacy for their STEM-TPACK developed significantly, with large effect sizes, after they co-designed the websites. The results also indicate that the preservice teachers faced challenges in communicating their discipline-based content knowledge when developing the STEM projects. Contextualizing and connecting their content knowledge with real-world design challenges was also difficult for them. Consequently, the preservice teachers realized that teaching is a complex matter, especially when they need to integrate the different disciplines for STEM education. However, this was viewed in a positive light for professional development. This study implies that preservice teachers may benefit from learning by design, employing the TPACK framework in the social setting of interdisciplinary STEM communities.

**Keywords:** preservice teachers; STEM education; technological pedagogical content knowledge

## 1. Introduction

The rapid development of science and technology has, in many ways, changed the demands of teachers' competencies. Two key competencies that preservice teachers should begin to develop include ICT integration for subject matter learning and the interdisciplinary teaching of Science, Technology, Engineering, and Mathematics (STEM) [1,2]. Teachers' competencies in ICT integration and facilitating interdisciplinary STEM-based learning are both likely to enhance students' knowledge and skills that are crucial to their career prospects [3]. Emerging research findings have also supported that pedagogically well-designed STEM activities can raise students' intention to learn, promote highly integrated understanding, and empower students for the 21st century workplace [4–6]. Falloon, Hatzigianni, and Bower advocated that STEM education should equip the young generation with STEM technological and design capabilities; skills; and the disposition to work productively and ethically in dynamic, complex, and challenging environments [7].

However, teachers are facing multiple challenges in teaching interdisciplinary STEM (ISTEM) education. First, teachers lack the necessary technological and engineering knowledge and skills to design and implement STEM curriculum [1,8–10]. Furthermore, STEM involves interdisciplinary curriculum integration, which is always challenging for schools and teachers to implement, given that schools are operationalized based on single-subject teaching [11–14]. Thus, ISTEM education faces multiple barriers, including time, interdisciplinary communication, teaching materials, and alignment to the curriculum standards [9]. For teachers to develop the capacities to facilitate ISTEM learning, it has been advocated that they should acquire substantial co-design experience involving diverse expertise in professional learning community to create the knowledge needed [15,16]. This study draws on the existing literature of the technological pedagogical content knowledge (TPACK) [17] and the notion of Webquest [18] to create a preservice teacher professional development (TPD) course to enhance their TPACK for STEM education. By operationalizing TPACK in the form of an updated Webquest [18], the study created a worked example of a STEM-TPACK Google site as a usable lesson design in which resources, applications, and activities can be developed and distributed to facilitate students' interdisciplinary STEM learning. For convenience in this study, the website is referred to as STEMQuest. The study investigates preservice teachers' changes in their STEM-related TPACK efficacies and their experiences of interdisciplinary collaboration to co-design the STEMQuest. It contributes to the current research on STEM education by drawing on the notion of TPACK as a possible framework to map out the knowledge that the preservice teachers need. While the TPACK framework has been employed in the context of STEM education research [1,19], whether and how it may foster preservice teachers' capacity for ICT integration and STEM education has apparently not been reported. Given preservice teachers' crowded curriculum, a study addressing these two teacher capacities may offer new insights into the restructuring of sustainable teacher education. Consequently, when teachers are able to foster students' STEM capacities, a sustainable growth in technologies and the economy can be achieved in the long run. Two research questions were formulated to guide this research:

- Are there significant differences in preservice teachers' STEM-TPACK efficacies after they complete the STEMQuest in the cross-disciplinary co-design team?
- How do preservice teachers describe their experiences in co-designing the STEMQuest?

## 2. Literature Review

### 2.1. Learning by Designing to Develop Teachers' TPACK

The TPACK framework has been introduced as a conceptual framework for researchers and educators to analyze the forms of fundamental knowledge (i.e., technological knowledge, pedagogical knowledge, and content knowledge) that the teachers need and the interrelated forms of knowledge they develop as they construct ICT-integrated lesson plans [17]. The past research indicates that the TPACK framework can guide teachers to design ICT-integrated lessons by scaffolding teachers' design thinking [20–24]. The studies, to date, provide substantial evidence that learning by design in a group has emerged as a reliable pedagogical approach to foster teachers' TPACK development. Nonetheless, the integration and development of multiple forms of knowledge in pedagogy, subject matter, and technology are cognitively demanding. Valtonen et al. highlighted that preservice teachers had concerns about their pedagogical knowledge and technological pedagogical knowledge, especially among first-year preservice teachers [25]. So and Kim, on the other hand, reported that preservice teachers found it difficult to craft authentic problems to engender student-centric learning [26]. In addition, Hughes, Cheah, Shi, and Hsiao's analyses indicated that preservice teachers were prone to using ICT in a more teacher-centric manner rather than adopting a student-centered approach for knowledge construction [27]. However, they lacked the required knowledge and skills. Pringle, Dawson, and Ritzhaupt's analyses of teachers' lesson plans indicated that science teachers might use more advanced technologies after TPD activities, but authentic science inquiry may not be evident in their lesson plans [28]. Koh's study revealed the complex design reasoning processes

when mathematics teachers engaged in the development of technological pedagogical mathematics knowledge (TPMK) [29].

In sum, the current literature indicates that teacher educators need to foster preservice teachers' capacities to design student-centric, ICT-integrated lessons that employ technologies as cognitive tools [27,30]. While learning by design has been identified as the approach to promoting teachers' competency in TPACK with several design models, as technology mapping and the "Scaffolded TPACK lesson design model" (STLDM) have articulated [20,31], additional instructional scaffoldings may be needed to foster teachers' TPACK. In addition, it has been pointed out that teachers' concerns about their lack of knowledge, lesson resources and activities, and nuanced implementation skills can only be resolved by engaging them in the iterative cycles of designing, implementing, and revising instruction [32]. In other words, most gaps in integrating ICT into classroom teaching are dependent on teachers' trajectory of learning by doing.

The idea of employing TPACK for the study of STEM education has been proposed and tested in limited contexts. Parker et al. investigated science teachers' quality of technology use with reference to the Next Generation Science Standards and found that high-quality use was limited [19]. Yildrim and Sidekli employed the TPACK framework in preservice teacher development by integrating technology in teaching mathematics in the context of STEM education [33]. They concluded that the STEM applications positively affected the teachers' mathematic literacy and TPACK development. Both studies were set in the context of STEM education, and the integrated use of technology is limited to single-subject content learning. As for the teacher use of ICT in engineering, very little is known, except that teachers' lack of engineering knowledge is an identified barrier for the implementation of STEM education [34–36]. Nonetheless, quantitative studies of teachers' technological pedagogical science, mathematics, and engineering knowledge influencing their interdisciplinary STEM efficacies have been recently reported [31,37]. These structural equational model and regression studies point to the importance of anchoring instruction in interdisciplinary STEM challenges and the reciprocal relationships between TPACK and STEM. Development in TPACK may promote STEM education and vice versa, which needs to be empirically supported by future studies.

This study draws on previous models that support teachers' lesson design for TPACK to develop the even more demanding STEM-TPACK, which, to date, has rarely been explored. It can extend the TPACK framework beyond a single subject.

## 2.2. Interdisciplinary STEM and TPACK

The reciprocal relationships among STEM subjects are well understood by any practicing engineers and well-informed learners. Simply put, science and mathematics advance through inductive and deductive thinking as basic research. Engineers apply the knowledge created by science and mathematics to create technologies that solve real-world problems through design thinking [38]. A case in point is the study of hydroponics as a form of food production technology. The technology is designed based on biological knowledge and the relevant computation of the nutrients needed for optimal yield [14]. The new technologies created may at times provide more efficient means for the observation or measurement of scientific properties or speedy computation for mathematics research, which creates new knowledge that facilitates engineering design. These disciplines intertwine to drive advancement in knowledge and technologies in a dynamic manner.

As the common element of both TPACK and STEM education, the role of technology may be different in different contexts. In the school context, it is less likely to be a created outcome of STEM, but may possibly be the pedagogical tool used to support teaching and learning [31]. Thus, when teachers are designing lessons for subject matter learning, they need to draw on their existing TPACK [39]. Nonetheless, when STEM education is anchored with engineering design challenges, students will encounter authentic engineering problems, which will require them to use ICT as a productive, collaborative, and cognitive tool to help them gather, synthesize, model, and construct the knowledge needed [30]. This could address the problems reported earlier about teachers not using ICT

as a cognitive tool to engender 21st century student-centered learning [19,27,28]. In addition, the latest advanced technologies, such as 3D printing, micro-computers, and block-based coding environments afford teachers and students ample opportunities to compute and make functional prototypes [40]. Such technologies for making need to be integrated in a relevant school curriculum for STEM education. Halverson and Peppler highlighted that making, coding, collaborating, and tinkering can act as an antidote to traditional schooling that is based on fixed curriculum goals, well-defined problems, and problem solving recipes [41]. Thus, STEM education may recast the problem of technology integration for teaching to technologies for learning and knowledge construction. This would provide a meaningful context for preservice teachers to make better sense of how technologies support teaching and learning, hence enhancing their TPACK. Similarly, this would also help students to make better sense of learning with technologies and build among them a habit of using technology to solve problems [42].

It is now commonly recognized that teachers need to develop TPACK to integrate technology into teaching, and it seems likely that STEM education would require teachers to activate and expand their TPACK for STEM lesson design. In addition, TPACK and STEM are both crucial to developing students' 21st century capacities [17,43]. Parker et al. associated teachers' TPACK with STEM and argued that these two fields of study need to be integrated [19]. While the blend of the two areas involves five elementary forms of knowledge (STEM subject knowledge plus pedagogical knowledge) that will interact in a complex manner, the foregoing argument shows that the concurrent development could be an outcome. Nonetheless, it also seems necessary to draw on the notion of distributed expertise to minimize the possibility of cognitive overload [44]. Therefore, it is necessary to provide support in the process of knowledge construction that incorporates distributed expertise through the use of the selected collaborative platform to mediate the emerging shared cognition [45].

### 2.3. Webquest as a Technological Pedagogical Tool

Dodge created a Webquest to guide students' web-based inquiry [18]. A Webquest enables students to use information drawn from the Internet to support higher-order thinking, such as the critical and creative use of information [46]. It can be regarded as a technological pedagogical model, with the Internet and inquiry-based learning, respectively, as the technological component and the pedagogical underpinning. Integrated with the content knowledge, Webquests could be one form of online TPACK that is relatively easy to master. The current research on Webquests indicates that the inquiry-oriented use of information coupled with cooperative learning strategies may enhance the learning outcomes of science, mathematics, and humanities subjects. Webquests are also welcomed by students and teachers [47–49]. However, Webquests seem to be less commonly used to enhance teachers' TPACK. Chuang reported that preservice teachers' beliefs about students may influence their design of Webquests, which may fail to engage students in higher-order learning [50].

Given the potential benefits of Webquests for students' learning and its multidiscipline applicability, this study expanded the notion of Webquests to create STEMQuests using the Google site. Research on teachers' use of the Google site has been emerging in recent years. The Google site has been reported to facilitate the sharing of resources among preservice teachers [51]. Wang, Jeng, and Huang employed the technology acceptance model to study teachers' continuous intention to use Google sites for collaboration [52]. The findings indicated that teachers' continual use is dependent on their perception of the ease of use and usefulness of Google sites. There are currently two studies about building Webquests on Google sites within the Web of Science database [52,53]. Given the current capacity of the Google site, which allows the easy embedding of applications and video resources, it offers many pedagogical affordances that could facilitate preservice TPD for STEM and TPACK. This study thus explores the effects of creating STEMQuests based on preservice teachers' technological pedagogical science knowledge (TPSK), technological pedagogical mathematics knowledge (TPMK), technological pedagogical engineering knowledge (TPEK), and ISTEM efficacy.

## 3. Method

This study adopted a mixed-methods approach to answer the research questions. A single group pre-posttest was employed to answer research question 1, which pertains to the effects of creating STEMQuests based on preservice teachers' efficacies with TPSK, TPMK, TPEK, and ISTEM. Follow-up interviews were arranged to understand the preservice teachers' experiences in this undertaking within the interdisciplinary co-design setting.

### 3.1. Background of Indonesia Teacher Education and Participants

This research was conducted in Indonesia. Based on the current policy, teachers in Indonesia have to first obtain a Bachelor's degree in education from an education university in order to be qualified for studying a one-year post-graduate teacher professional training program. Upon the completion of the professional training, teachers will earn certification to teach in public schools. In the four-year undergraduate program, the preservice teachers study specialized subjects, such as science, mathematics, language, and engineering. The participants of this study were pre-service teachers from the teacher education programme in the second author's university. They did not have prior experiences in integrated STEM learning.

Thirty-seven second- and third-year preservice teachers from the science (N = 17, Male = 4), mathematics (N = 8, Male = 2), computer science (N = 6, Male = 6), and engineering (N = 6, Male = 4) department were invited to participate in this study voluntarily. The average age of the preservice teachers was 22.1 (standard deviation = 3.31). In total, there were 21 female teachers and 16 male teachers. They were recruited to participate in this study to represent the science, math, and engineering departments (computer science and engineering are both regarded as an engineering major). Before the workshop, the preservice teachers were grouped into eight groups, with at least one teacher from the science, mathematics, and engineering disciplines.

### 3.2. The STEMQuest Design-Based Learning for STEM-TPACK Development

The first and second author were the instructors of the STEM-TPACK workshop. The first author is in the field of educational technology and teacher education, while the second author is in the field of science education, specializing in chemistry. The STEMQuest design-based learning workshop with follow-up group-based design activities lasted for one month. It was initiated with two full-day workshops, followed by weekly discussions, tweaking, presentations, peer reviews, and the iterative refinement of the website. The first day of the workshop was a two-hour lecture about STEM, TPACK, and Webquests. The lectures provided the basic theoretical and pedagogical knowledge needed for the preservice teachers. A worked example of STEMQuests (e.g., https://sites.google.com/view/making-a-musical-instrument/home) was provided for the preservice teachers to browse through.

The worked example site can be duplicated by enrolling the preservice teachers as the editors of the site. The preservice teachers can build on the structure to choose new design challenges and perform the necessary content and task analyses. The TPD processes along with the pedagogical structure of the STEMQuest are illustrated in Figure 1. The pedagogical function of each page was explained to the preservice teachers to understand the what and why of the design. In essence, the STEMQuest was anchored with an engineering design challenge. The collaborative works were directed by rules and role specification. The self-directed component was guided by the evaluation rubrics and post-design reflection. One to two webpages were created specifically for each of the STEM subjects, providing instructional resources (e.g., instructional videos), scaffoldings (spreadsheets or infographics), and directions for the use of possible applications as inquiry or cognitive tools (e.g., Audacity, to review sound wave patterns). For example, the webpages of mathematics usually require students to articulate the quantitative variables that need to be measured. Spreadsheet files or other ICT applications can be embedded for students to access. The formulas connecting the variables can

be inserted in the relevant cells in the spreadsheet or scripted on other ICT applications. This allows the students to view the outputs as the data are being inputted. Alternatively, the teachers can request the students to write or script the formulas.

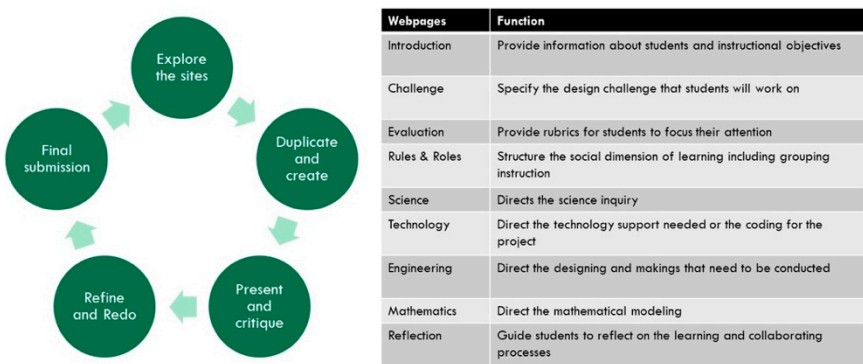

**Figure 1.** Teacher Professional Development (TPD) processes and basic structure of a STEMQuest.

The preservice teachers duplicated the worked example site and started to discuss the design challenge for the group. They then divided the labor based on their expertise. The stages of development included the hands-on building of the website; researching, refining, and developing teaching materials; periodic discussion to check on the progress and exchange ideas. The first draft of the design was completed at the end of the workshop, followed by the rest of the iterative design activities, as shown in Figure 1.

*3.3. Data Collection*

The quantitative data collection was conducted through a pre-post survey. The qualitative data collection through interviews, reflective journals, and observation was carried out throughout the STEMQuest design processes.

The Technological Pedagogical STEM Knowledge (TP-STEMPK) survey was validated by Chai et al. [31] to measure the teachers' self-efficacies in TPSK, TPMK, TPEK, and ISTEM. Kelly, Knowles, Holland, and Han argued that teachers' efficacies were the important factors in their instructional practices and hence in students' learning outcomes [54]. They advocated that the measurement of self-efficacy should be contextualized. Based on the TPACK framework, this study employed the TP-STEMPK survey to collect the pre-post test data from the teachers. The TP-STEMPK measures teachers' efficacy and it includes four factors—namely, teachers' TPSK (4 items), TPMK (5 items), TPEK (4 items), and ISTEM (4 items). The scale is rated by a 7-point Likert Scale (1 for "strongly disagree" and 7 for "strongly agree"). Exploratory and confirmatory factor analyses were conducted to ascertain the construct validity and that the original instrument has adequate convergent and discriminant validity [31]. The items were translated to Bahasa Indonesia by the second author and independently checked by another Indonesian lecturer for accuracy and readability. A bilingual version was presented to the teacher participants to ensure they could clearly understand the meaning of the items. The alpha reliabilities for the validation study and a sample item are provided, respectively, as follows:

TPSK (0.87; 0.86): I am competent in helping my students to critically synthesize information from various web-based resources for science investigation.

TPMK (0.89; 0.82): I can engage students in constructing possible mathematical models about real-world problems with appropriate technologies (e.g., spreadsheet).

TPEK (0.89; 0.84): I am competent in facilitating students' learning of various software tools that engineers use to develop their ideas (e.g., computer-assisted design tools).

ISTEM (0.91; 0.93): I can design lessons that appropriately integrate interdisciplinary STEM content and technology for student-centered learning.

The preservice teachers were observed during the co-design processes. The observation focused on the interaction patterns and conversations of the group discussions. The preservice teachers also wrote reflective journals regarding their experiences. The stimulated questions in the reflective journals are:

- What do you think about the workshop?
- What is your understanding about STEM? How did you learn in the workshop?
- What is your understanding about TPACK?
- How do you find the project idea and design?
- What do you think about the group work in this workshop?
- What are the knowledge and skills that you developed?

After the final submission, the preservice teachers were interviewed with the following questions:

- What are the challenges you faced in developing the collaborative interdisciplinary STEM project?
- What are the strategies for designing lessons with a TPACK framework?

The questions for the reflective journals and the interviews were formulated to solicit the preservice teachers' experience of STEMQuest making. They provide a means for the researchers to gain a deeper understanding of the preservice teachers' perspectives. The questions were as open-ended as possible to prevent bias in the findings.

### 3.4. Data Analysis

The quantitative items were first checked to see if the requirement of normality was met. Since some of the differences between the pre-post scores had Kurtosis scores greater than 2, which indicated that the data did not meet the requirement for normality, a non-parametric test was employed. The quantitative data were analyzed with SPSS version 25. Reliability analyses were conducted and the mean of each factor was computed. The Wilcoxon signed-rank test was then run and the Pearson correlation coefficient r was computed to estimate the effect sizes. The qualitative data from the transcribed interviews and students' reflective journals were analyzed through an inductive analysis to derive salient themes about the preservice teachers' experiences. The observation data were used to triangulate the emerging themes. The research members cross-checked their data interpretation with the participants for accuracy.

### 4. Result and Discussion

This section presents the findings, followed by a discussion to articulate the connections between the literature and the findings.

### 4.1. Are There Significant Differences in the Preservice Teachers' STEM-TPACK Efficacies after They Complete the STEMQuest in the Cross-Disciplinary Co-Design Team?

Table 1 shows the Wilcoxon signed-rank test results before and after the preservice teachers' co-design experience in STEMQuests. The findings indicated that the preservice teachers' efficacy in the four factors (i.e., TPSK, TPEK, TPMK, and STEM) changed significantly, and the effect sizes for the change were in the range of large effect sizes (r > 0.50).

The past studies provided substantial evidence that engaging preservice teachers in the learning by design approach with a TPACK framework could enhance their capacity to design ICT-integrated lessons [20–24]. While the relevant literature has been confined to a single subject or the general TPACK efficacy, this study focused on the importance of distributed expertise [44] by grouping the preservice teachers from different disciplines together. The meaningful context of the preservice teachers' self-selected engineering challenges was likely to reduce the effect of cognitive overload since it avoided meaningless, disconnected or personally disengaged learning. It is therefore important for teacher educators to further experiment with the idea of enhancing TPACK through STEM.

Drawing on the notion of and past research on Webquests [18,48,49,55], STEMQuests extended the notion of Webquests as an interdisciplinary pedagogical framework. It mediates and coordinates the distributed expertise through a collaborative platform showing the emerging shared cognition [45] of all co-designers. This may mitigate coordination and communication problems common to STEM reforms [9]. The STEMQuests built on the Google site not only facilitated the sharing of resources among the preservice teachers [51], but also further reinforced the sharing and growing of expertise as suggested by the findings. Wang et al. proposed that teachers' continual intention to use the Google sites is dependent on their perceived usefulness of the technology [52]. This study points to the importance of designing good pedagogical use of the Google site in the form of STEMQuests. Future research may look into whether creating STEMQuests or Webquests could change teachers' perceived usefulness of the Google site and hence strengthen teachers' continual intention to learn and use similar collaborative authoring platforms.

**Table 1.** Wilcoxon signed-ranks test.

|  |  | Mean | Std. Deviation | Wilcoxon-Test | r |
|---|---|---|---|---|---|
| TPSK | Post | 5.45 | 0.63 | 4.51 *** | 0.56 |
|  | Pre | 4.13 | 1.14 |  |  |
| TPEK | Post | 5.21 | 0.71 | 4.46 *** | 0.55 |
|  | Pre | 4.07 | 1.00 |  |  |
| TPMK | Post | 5.31 | 0.85 | 4.38 *** | 0.54 |
|  | Pre | 4.19 | 0.95 |  |  |
| STEM | Post | 5.51 | 0.69 | 4.72 *** | 0.58 |
|  | Pre | 3.97 | 1.31 |  |  |

Note 1. *** $p < 0.001$.

*4.2. How Do Preservice Teachers Describe Their Experiences in Co-Designing the STEMQuest?*

Three themes emerged through the inductive analysis of the verbal data collected. Some excerpts are shown in the following sections.

4.2.1. STEM as Interdisciplinary Authentic Learning

In teacher education programs, preservice teachers often learn the subject as an isolated discipline rather than in an interdisciplinary context. This interdisciplinary project provided ample opportunities for the preservice teachers in making connections of all the STEM subjects and the authentic applications of the subject matters to real-world problems. The interdisciplinary context challenged them to construct new knowledge, especially in applying them, as reflected by the quotes below.

*"We started by finding the problems and identifying the aspects that can be affected by the problems, followed by browsing and asking questions to integrate all STEM disciplines." (Preservice teacher 9, Interview, 10 May 2018)*

*"I tried to understand the relation among the characteristics of chemistry knowledge in the project of building materials that we need to develop. It seems that deeper knowledge is required to understand the relationship among the chemistry knowledge." (Preservice teacher 14, Interview, 10 May 2018)*

*"I found that the learning experiences have challenged me to understand different basic principles of each discipline and their connections. I realized that we have to understand the multidisciplinary approach to solving the daily life problems." (Preservice teacher 8, Interview, 10 May 2018)*

*"The experiences have challenged my thinking to apply the knowledge and make connections, besides considering students' responses when it is implemented in the classroom." (Preservice teacher 5, Interview, 10 May 2018)*

As the preservice teachers were developing the STEMQuests, they learnt to contribute their respective content knowledge based on the chosen design challenge and negotiate to resolve the challenges, such as finding a suitable classroom context for the STEMQuests. An example from the group who chose to conduct a "green building" project is illustrated below.

| | |
|---|---|
| Preservice teacher 1 | "I think the green building can be related to the topic in Biology. We learn about the plant that can absorb the pollutants and the insects that can act as the indicators of pollution." |
| Preservice teacher 2 | "It is also related to the chemistry concepts in pollutants. However, how can engineering and technology integrate in students' learning?" |
| Preservice teacher 3 | "Ehm … I am thinking of using the software in the building design. We have that subject in engineering." |
| Preservice teacher 4 | "But we don't have that subject in secondary schools." |
| Preservice teacher 5 | "We can implement it in vocational schools." |

This type of conversation was commonly observed in all the groups as they attempted to create the STEMQuests. While making connections was challenging, the preservice teachers found it meaningful, as the interdisciplinary learning provided opportunities for developing higher-order thinking skills and connections with other subject areas. Therefore, the preservice teachers' co-design experiences strengthened their capacities to design STEM projects that could foster interdisciplinary STEM learning [4–6].

Besides this, the preservice teachers also thought about how to make the subject connections clear to students because they rarely engaged in interdisciplinary learning. The teachers developed ideas that are relevant to students' daily lives and the Indonesian context to create authentic learning experiences for them to understand STEM teaching as technology-enabled 21st century learning practices [56].

### 4.2.2. TPACK as a Complex Student Learning-Focused Design

The topics of the eight STEMQuests created by the preservice teachers are an exoskeleton, a wind-turbine generator, building materials, green building, piezoelectric, a nano-generator, fruits as alternative energy, and recycled plastic waste. The products indicated that the preservice teachers were able to re-create functional Google sites with the worked example. All the Google sites included student-centered learning activities, such as the creation of electronic concept maps, modelling relationships with spreadsheets and GeoGebra, and sketching with computer-based sketching tools. The existing TPACK research indicated that adopting technologies to promote student-centered inquiry-based science teaching is not a common practice [19,27,28]. Drawing on Webquests [18], STEMQuests are able to move towards student-centered learning of STEM. More importantly, after the experience of co-designing the STEMQuests, the preservice teachers realized that classroom teaching was complex, with many contextual uncertainties to deal with. They were constantly thinking about students' learning, which is important for the design of student-centered learning.

*"It was complicated for me as I realized the complex process in designing the lesson. I have to think about the content of STEM and students' learning experiences of classroom activities in detail." (Preservice teacher 13, Interview, 10 May 2018)*

*"It is important to understand students' characteristics, as we will give them the challenges to solve the problems and to make connections of their knowledge." (Preservice teacher 25, Reflective Journal, 9 May 2018)*

*"We need to design the problems which are relevant to students' daily lives, so that the students can make sense of the STEM projects as they generate the solutions." (Preservice teacher 33, Reflective Journal, 9 May 2018)*

The emphasis on students' learning was also observed in the preservice teachers' discussion.

| Preservice teacher 1 | "I think we can challenge students' thinking about designing the building." |
| Preservice teacher 2 | "I agree, but what is the relevance to other subjects?" |
| Preservice teacher 3 | "Can we challenge students' thinking about what kind of building materials from a chemistry perspective?" |
| Preservice teacher 4 | "Yes, we can, but we need to understand what students' prior knowledge is." |
| Preservice teacher 5 | "Yes, I agree." |

The work of creating a STEMQuest, as illustrated here, involves complex reasoning with heavy reference to students, who were the target users of the design. It is essentially an effort to create contextualized knowledge [24,25].

### 4.2.3. Collaboration Skills in an Interdisciplinary Context

When developing the STEM projects, the preservice teachers learnt to collaborate and communicate with their peers from different disciplines. The quotes cited earlier have already provided some indications that connecting the different types of content knowledge and addressing the contextual concerns were challenging for the preservice teachers. The quotes below further illustrate the communication challenges they faced.

*"We have different discipline background. It was difficult to connect our disciplines because everyone would like to put more emphasis on their own disciplines. After discussing with the lecturer, we found it important to understand the STEM project and the TPACK framework. Finally, we can come up with the ideas after discussion." (Preservice teacher 15, Reflective Journal, 9 May 2018)*

*"As we come from different disciplines, we faced difficult challenges. Initially it was difficult to combine our different ideas because the project was about one of the fields of science. But after the consultation with the facilitator, and following the steps with the TPACK STEM principles, we managed to bring together our ideas and perceptions of our disciplines" (Preservice teacher 17, Reflective Journal, 9 May 2018)*

The preservice teachers come from different departments of the university. They encountered challenges in understanding the epistemic and pedagogical practices of other disciplines. They were mindful not to dominate the STEM project with only their discipline. They tried to see things from the perspectives of other disciplines and communicate their ideas accordingly. Similar challenges were reported from practicing teachers' perspectives [9,11,12,14]. The preservice teachers' experiences could be good preparation for them before they enter the teaching service. The STEMQuest functions as a collaborative platform for them to work together and contribute the ideas, different resources, and technologies that they possess as part of their disciplinary knowledge.

## 5. Conclusions

This study began with the premise that teaching interdisciplinary STEM projects would bring about substantial development among teachers. Their key challenge was the multiple forms of knowledge that needed to be articulated, coordinated, synthesized, and transformed into concrete and feasible classroom activities that could engender meaningful learning among students [57]. The study drew on the professional learning community to address the importance of the distributed expertise and social capital [15,58], and expanded the TPACK literature to cater for the interdisciplinary education. Webquests [18] were redesigned as STEMQuests to epitomize the TPACK-STEM framework. The research outcomes indicated that STEMQuests could enhance preservice teachers' TPSK, TPMK, TPEK, and ISTEM concurrently, which is beyond expectations, as most TPACK research aims to enhance TPACK in a single subject. The mean scores of the post survey for the four factors were higher than those reported among Chinese teachers who attended some STEM training [31], though statistical significance could not be established. In addition, the experiences that the preservice

teachers went through seemed to launch them towards the trajectory of student-oriented lesson design and interdisciplinary sense making. While the existing studies on TPD for STEM may focus on aspects that are different from teachers' needs, such as changes in attitudes or beliefs [14,34,35], the STEMQuest approach, as depicted in this study, aimed at developing teachers' knowledge about using content-specific technology (i.e., TCK) and harnessing their capacity to design TPACK for various disciplines (science, mathematics, and engineering) to support students' STEM learning. The findings in this study indicated that the instructional model had large effects among the STEM-TPACK of Indonesian preservice teachers. Overall, the notion of STEMQuests may warrant further exploration among teachers elsewhere in the world.

There are several limitations in this study. First, this study adopted a single-group pre-test and post-test design, which is not a strong research design. Future research could employ a quasi-experimental approach to compare different ways of preparing a STEM curriculum (e.g., using a Wiki versus Google site) to understand the effects of the mediating platforms. Second, this study employed self-reported efficacies as the dependent data. While the quantitative outcome could be supported by the qualitative positive experiences reported by the preservice teachers, and the completed websites attested to their basic mastery, it may be necessary to include objective performance evaluation [59]. Third, this study did not measure the preservice teachers' communication skills, which may have influenced the outcome of this study. Future studies could look into this aspect and measure the relevant aspects of skill as a covariate in quantitative assessments. Finally, this study was premised on a meaningful learning-by-doing approach [30]. It requires the preservice teachers to have a conceptual change to truly master the power of "learning with technology", which the current study may not have achieved [60,61].

**Author Contributions:** Conceptualization, methodology, validation, analysis, C.S.C.; investigation, resources, qualitative data collection, and analyses, Y.R.; member checking, data triangulation, writing—original draft preparation, writing—review and editing, C.S.C., Y.R., and M.S.-Y.J. All authors have read and agreed to the published version of the manuscript.

**Funding:** This research received no external funding.

**Conflicts of Interest:** The authors declare no conflict of interest.

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
