# Peer review of "Indonesian Science, Mathematics, and Engineering Preservice Teachers’ Experiences in STEM-TPACK Design-Based Learning"

_sustainability, doi:10.3390/su12219050_

Round 1
Reviewer 1 Report
Excellent subject. The methodology and data analysis are adequate. However, it would be relevant to clarify the limits of learning by doing and meaningful learning, which requires a conceptual change in Posner and his collaborators' sense. Likewise, it would be relevant to specify the conditions, so preservice teachers integrate technology for mathematics in the context of STEM in education.
Author Response
please find the attached respond

Reviewer 2 Report
- This topic is interesting and provides the useful information for the teacher education.
- The focus of this study is the Indonesia preservice teachers. I suggest the author should provide more local information about the learning and professional situations of these preschool teachers in Indonesia.
- The section of the method is well-written. I suggest the author could provide more learning backgrounds or prior educational experiences about STEM for the participants.
- In the section of the conclusion, I suggest the author could provide the comparative analysis between Indonesia and other countries about this topic and focus on the advantages of this instructional model in Indonesia teacher education.
Author Response
please find the attached respond

Reviewer 3 Report
- many English mistakes, English should be checked once more, e.g., line 27 "This study implies that teacher education curricula may benefits" => "This study implies that teacher education curricula may benefit", line 68 "Consequently, when teachers are able foster students’ STEM capacities" => "Consequently, when teachers are able to foster students’ STEM capacities", line 71 "Are there significance differences in the preservice" => "Are there significant differences in the preservice", line 211 "The pedagogical function for each page were explained" =>"The pedagogical function for each page was explained", line 250 "The items were translated to Bahasa Indonesia and checked by Indonesia Lecturer" => "The items were translated into Bahasa Indonesian and checked by an Indonesian Lecturer", line 271 "the preservice teachers were interview" =>the preservice teachers were interviewed", lines 301-302 "The finding that getting preservice teachers with diverse background to co-design the STEMQuest can concurrently enhances" => "The finding that getting preservice 301 teachers with diverse background to co-design the STEMQuest can concurrently enhance", lines 319-320 "the Google site not only facilitated in sharing of resources" => "the Google site not only facilitated the sharing of resources", line 328 "analysis of the verbal date" => "analysis of the verbal data", line 366 "as the interdisciplinary learning provide opportunities for developing" => "as the interdisciplinary learning provides opportunities for developing", line 370 "Besides the preservice teachers also thought about how to making connection with" => "Besides the preservice teachers also thought about how to make a connection with", line 402 "relevant with" => "relevant to", line 407 "In the conversations show that pre-service teachers understand" => "In the conversations it was shown that pre-service teachers understand"
- check punctuation, for example, in line 206 "(e.g. https://sites.google.com/view/making-a-musical-instrument/home)", comma after e.g., in line 235, 21 should not be placed in brackets "There were (21) female teachers and 16 male teachers", in line 370 "Besides the preservice teachers also thought", a comma after besides would help the reader in decoding the message
- a link given in the article could not be accessed “(e.g. https://sites.google.com/view/making-a-musical-instrument/home)” (line 206)
- all the acronyms should be explained when they appear for the first time, otherwise it is difficult to follow the topic development
- in line 250, I could not see the relevance of the sentence "The items were translated to Bahasa Indonesia and checked by Indonesia Lecturer", it should be explained, and the language mistakes should be corrected
- in lines 353-354, 410-411, 412-413, and 414-448 (the entire sub-section 4.2.3), the information is not clearly understood, particularly because of the language use/mistakes
Author Response
Please find the attached responds

Reviewer 4 Report
The subject of the manuscript is relevant and current. However, different changes are proposed, which are detailed below.
- On the second page, interdisciplinary STEM is referred to in two ways: I-STEM (line 45) and iSTEM (line 51). The same initials must always be used. Review this question throughout the text.
- Page 2, line 85. The following must be correctly cited within the text: Valtonen, Leppänen, Hyypiä, et.al
- Page 2, line 89. The symbol “&” must be replaced by “and” in Hughes, Cheah, Shi & Hsiao
- Page 4, line 168. Include [18] at the end of the first sentence.
- Page 4, line 183. The authors indicated do not coincide with those included in Reference 14.
- Page 5. Within the Method section, the subsection related to Participants should appear earlier in the text than the one called The STEMQuest Design-Based Learning for STEM-TPACK Development
- The number of preservice teachers must be specified for each of the specialties, that is, science, mathematics, computer science, and engineering.
- There should be a greater description of both the preservice teachers and the teachers. Thus, for example, in the case of the former, indicate distribution by gender, age, etc.
- Data Collection. It is necessary to justify the questions that make up both the reflective journal and the interview.
- Data Analysis. Given that the sample size is not very large, it must be indicated whether the data met the requirements of normality and homogeneity of variance to carry out parametric quantitative data analysis tests (Paired sample t-tests). In the event that they are not met, it is recommended to use non-parametric tests (specifically, the Wilcoxon test).
- Table 1. In the first column you should specify, instead of Pair 1, Pair 2, etc., TPSK, TPEK, TPMK, and STEM. Also, the second column should include Pre and then Post. Moreover, the significance level (p) must be specified.
- Verbal extracts from the participants are quoted in section 4.2. It is recommended not to use quotes and italics together.
- References section:
Reference 3. The final page of the article is missing.
Reference 26. The initial and final pages of the article are missing.
Reference 36. Delete the article pages and indicate “Advance online publication”.
Reference 37. The final page of the article is missing.
Reference 40. The book chapter must be cited correctly.
Reference 47. Delete the article pages and indicate “Advance online publication”.
Reference 51. The initial and final pages of the article are missing.
Author Response
Please find the attached responds

Round 2
Reviewer 3 Report
Dear Authors,
The second submitted version is much clearer than the first one. Only the English should be checked once more as there are some revisions that still need be made.
Kind regards,
